# Vitamin D Status and Parkinson’s Disease

**DOI:** 10.3390/brainsci12060790

**Published:** 2022-06-16

**Authors:** Michela Barichella, Federica Garrì, Serena Caronni, Carlotta Bolliri, Luciano Zocchi, Maria Carmela Macchione, Valentina Ferri, Daniela Calandrella, Gianni Pezzoli

**Affiliations:** 1Parkinson Institute, ASST Gaetano Pini-CTO, 20126 Milan, Italy; barichella@parkinson.it; 2Fondazione Grigioni per il Morbo di Parkinson, 20126 Milan, Italy; caronni@parkinson.it (S.C.); bolliri@parkinson.it (C.B.); mariacarmela.macchione@gmail.com (M.C.M.); ferri@parkinson.it (V.F.); calandrella@parkinson.it (D.C.); pezzoli@parkinson.it (G.P.); 3Dipartimento di Fisiopatologia Medico-Chirurgica e dei Trapianti, Università degli Studi di Milano, Sezione di Fisiologia, 20133 Milan, Italy; luciano.zocchi@unimi.it

**Keywords:** vitamin D, vitamin D insufficiency, vitamin D deficiency, hypovitaminosis D, Parkinson’s disease, vitamin D receptor (VDR) polymorphisms

## Abstract

Parkinson’s disease (PD) is a complex and progressive neurodegenerative disease, characterized by resting tremor, rigidity, slowness of movement, and postural instability. Furthermore, PD is associated with a wide spectrum of non-motor symptoms that add to overall disability. In recent years, some investigations, from basic science to clinical applications, have focused on the role of vitamin D in PD, often with controversial findings. Vitamin D has widespread effects on several biological processes in the central nervous system, including neurotransmission in dopaminergic neural circuits. Various studies have recorded lower levels of vitamin D in PD patients than in healthy controls. Low vitamin D status has also been correlated with the risk for PD and motor severity, whereas less is known about the effects vitamin D has on cognitive function and other non-motor symptoms. This review aims to better characterize the correlation between vitamin D and PD, clarify the role of vitamin D in PD prevention and treatment, and discuss avenues for future research in this field.

## 1. Introduction

Parkinson’s Disease (PD) is the second most common degenerative disease of the central nervous system (CNS) after Alzheimer’s Disease (AD). PD is a complex and progressive neurodegenerative disease where motor symptoms such as bradykinesia, resting tremor, rigidity, postural instability, and gait impairment are the most prominent features. PD is also characterized by a wide variety of non-motor symptoms (hyposmia, constipation, urinary dysfunction, orthostatic hypotension, cognitive impairment, depression, and rapid eye movement sleep behavioral disorder), that add to overall disability. These non-motor symptoms may present several years or even decades before the occurrence of motor features [1]. The pathophysiology of PD is characterized by the presence of intraneuronal cytoplasmic inclusions in the CNS, mostly consisting of alpha synuclein aggregates, known as Lewy bodies, which lead to dopaminergic loss in the subtantia nigra pars compacta (SNpc) and other nuclei [2]. The underlying molecular pathogenesis involves α-synuclein proteostasis along with other pathways and mechanisms: mitochondrial function, oxidative stress, calcium homeostasis, axonal transport, and neuroinflammation [3]. Although the pathophysiology of PD is well characterized, its exact etiology still remains elusive. Among several etiological factors, low vitamin D status has recently emerged as a possible modifiable risk factor for PD.

## 2. Methods

A Medline review of publications from 2010 to January 2022 was conducted using a combination of words and MeSH terms: “vitamin D”, “vitamin D insufficiency”, “vitamin D deficiency”, “hypovitaminosis D”, “Parkinson’s disease”, “Parkinson’s disease risk”, “vitamin D receptor (VDR) polymorphisms”, “VDR”, and “COVID-19”. Only articles written in English were selected for study. A clinicaltrial.gov search of concluded and ongoing studies on vitamin D and PD was also carried out.

## 3. Vitamin D Sources and Metabolism

Vitamin D is a fat-soluble hormone which is mainly synthesized in the skin from 7-dehydrocholesterol through the action of ultraviolet B radiation; however, vitamin D can also be obtained from diet. Vitamin D3 is biologically inert and must undergo two rounds of hydroxylation to become active. The first hydroxylation occurs in the liver by vitamin D-25-hydoxylase (possibly encoded by CYP2R1) to generate the circulating form of vitamin D: 25-hydroxyvitamin D3 (25-OH-D3) or calcidiol. Later, 25-OH-D3 is converted into the active hormone, 1,25-hydroxyvitamin D3 (1,25-(OH)_2_D_3_), also known as calcitriol, by 25-hydroxyvitamin D-1α-hydroxylase (CYP27B1) or 1α-hydroxylase. This second step occurs in the kidney, specifically in proximal convoluted tubule cells, and is tightly regulated by blood calcium and phosphorus levels [4]. Vitamin D metabolism is highly regulated. In fact, in response to low serum calcium, parathyroid hormone (PTH), secreted by the parathyroid glands, stimulates the transcription of CYP27B1, promoting production of 1,25-(OH)_2_D_3_; moreover, low blood levels of phosphate stimulate the expression of CYP27B1, promoting conversion of 25-OH-D3 into its active form. Active vitamin D3 acts on the small intestine, stimulating the absorption of dietary calcium and phosphate; in addition, it promotes, along with PTH, reabsorption of calcium in the distal renal tubule [5]. In addition to the kidneys, other cells and tissues are able to express CYP27B1, allowing the synthesis of its own 1,25-(OH)_2_D_3_ at a local level. In other tissues, CYP27B1 expression is primarily regulated by cytokines such as TNF and IFN-γ. By contrast, 1,25-(OH)_2_D_3_ regulates its own levels by inhibition of CYP27B1. Furthermore, 1,25-(OH)_2_D_3_ indirectly regulates its own levels via CYP24A1, a 24-hydroxylase, which has a key role in the catabolism of both 25-OH-D_3_ and 1,25-(OH)_2_D_3_ [6]. Vitamin D can also be obtained from some dietary sources. Two forms of dietary vitamin D exist in nature: vitamin D3 (25-OH-D_3_) or cholecalciferol which derives from the consumption of some foods of animal origin (oily or bluefish, egg yolk, and meat are particularly rich in vitamin D3) and vitamin D_2_ (25-OH-D_2_) or ergocalciferol, which is mainly contained in nuts (almonds, walnuts), mushrooms, beans, and green leafy vegetables. Similar to cholecalciferol, ergocalciferol must be hydroxylated at positions 25 and 1α to become maximally active.

The sum of cholecalciferol and ergocalciferol is what is generally referred to 25-OH-D, as this metabolite is used to measure vitamin D status [7]. Vitamin D dietary intake is usually insufficient, since only some foods are able to provide a reasonable amount of vitamin D [8]. For this reason, many foods nowadays are enriched with vitamin D in order to ensure a good dietary intake. Moreover, several over-the-counter products contain vitamin D in different forms (capsule, tablets, liquids), combined with or without calcium [9].

## 4. Vitamin D Deficiency: Definition and Prevalence

Even if calcitriol is about 500–1000-fold more active than its precursor 25-OH-D, the latter is usually measured to estimate systemic vitamin D status. Several reasons justify the measurement of 25-OH-D instead of the active form. First of all, blood concentrations of calcitriol are extremely low. Moreover, 25-OH-D is more stable since it has a longer half-life (about 2–3 weeks) compared to the 1,25-(OH)_2_D metabolite (about 4–6 h) [10].

The definition of optimal vitamin D status is elusive since, to date, there is no evidence-based consensus on what levels of 25-OH-D define vitamin D deficiency.

The clinical practice guidelines of the Endocrine Society Task Force on Vitamin D [11] have established a cut-off level of 50 nmol/L (or 20 ng/mL) as **vitamin D deficiency**. A cut-off of <30 nmol/L (or 12 ng/mL) dramatically increases the risk of osteomalacia and nutritional rickets, and is therefore considered as **severe vitamin D deficiency** [11]. Concentrations of 25-OH-D between 50 nmol and 75 nmol/L (<30 ng/mL) should be considered as **vitamin D insufficiency**. Conversely, 25-OH-D concentrations between 75 and 150 nmol/L represent the normal range [11], whereas 25-OH-D concentrations between 150 and 250 nmol/L should be considered as increased levels, and concentrations greater than 250 nmol/L as overload [12].

In summary, serum vitamin D levels are commonly defined as deficient at <20 ng/mL, insufficient from 20 to 30 ng/mL, and sufficient at >30 ng/mL. A low vitamin D status, or hypovitaminosis D, comprises both deficiency and insufficiency, and is emerging as a very common condition worldwide. The prevalence of hypovitaminosis D differs according to the serum threshold used to define deficiency and is related to ethnicity, age and other factors. The prevalence of vitamin D deficiency ranges from 5.9% in the US [13] to 7.4% in Canada [14], and 13% in Europe [15], whereas the estimated prevalence of vitamin D insufficiency rises to 24% in the US [13], 37% in Canada [14], and 40% in Europe [15]. Moreover, lower levels of vitamin D are more common in childhood and the elderly. Notably, **severe vitamin D deficiency** occurs in patients affected by hepatic and/or renal failure, with a prevalence ranging from 85 to 99% [16].

## 5. Role of Vitamin D in Parkinson’s Disease

### 5.1. Vitamin D Status in Parkinson’s Disease

Several studies from basic science to clinical applications have highlighted a strong association between hypovitaminosis D and PD, even if no definitive conclusions are available and the literature has often shown controversial results (Table 1).

Hypovitaminosis D in PD patients, compared with healthy controls, has been reported. In 2011, Evatt et al. [17] reported a high prevalence of vitamin D insufficiency (concentrations lower than 30 ng/mL and greater than 20 ng/mL) and deficiency (concentrations lower than 20 ng/mL), respectively, in 69.4% and 26.1% of baseline samples collected from PD patients enrolled in the previous DATATOP study. DATATOP study included patients who had been diagnosed with idiopathic PD within the previous 5 years, and who presented mild symptoms which did not yet require medical therapy [18]. No decline in vitamin D levels was observed during the course of the disease, since after a mean follow-up time of 18.9 months, percentages of vitamin D insufficiency and deficiency decreased to 51.6% and 7.0%. A comparative study conducted on 300 individuals (100 patients with PD, 100 patients with AD, and 100 healthy controls) selected from the Clinical Research in Neurology database, confirmed a significantly higher prevalence of vitamin D insufficiency (concentrations lower than 30 ng/mL) in 55% of PD patients compared to both healthy controls (36%) and patients with AD (41%). Similarly, 23% of the PD patients had vitamin D deficiency (concentrations lower than 20 ng/mL) compared with 16% of the AD cohort and 10% of controls [19]. Another study, The Harvard Biomarker study [20], which enrolled 388 PD patients and 283 controls found that at least 17% of all PD patients had vitamin D deficiency (almost twice the prevalence in controls of similar ages). Furthermore, vitamin D insufficiency was reported in 47.2% of PD individuals. Further subgroup analyses highlighted strong and significant associations between vitamin D levels and PD in males, whereas in females concordant trends were observed without reaching statistical significance [20]. Insufficient vitamin D intake was reported in PD patients. One study, which focused on dietary intake in Belgian PD patients [21], revealed an inadequate intake of vitamin D in more than half of the enrolled patients; specifically, a mean daily intake of vitamin D below 10 µg (400 IU) was recorded in 62.5% of males and 40% of females.

**Table 1 brainsci-12-00790-t001:** Most relevant studies showing association of Vitamin D status with Parkinson’s Disease (PD). Legend of abbreviations: PD (Parkinson’s Disease); HC (Healthy Controls). * Patients from DATATOP: The Deprenyl and Tocopherol Antioxidative Therapy of Parkinsonism (DATATOP) cohort is a well-characterized cohort of subjects with early, nondisabling PD.

Association of Vitamin D Status with Parkinson’s Disease							
Ref.	Authorship	Type	Year of Publication	N° PD Patients	N° HC	Prevalence of Deficiency	Prevalence of Insuffiency
[17]	Evatt ML et al.	observational	2011	199 *	//	26.1%	69.4%
[19]	Evatt ML et al.	case-control study	2008	100	100	23% vs. 10%	55% (PD) vs. 36% (HC)
[20]	Ding H et al.	cross-sectional and longitudinal case-control study	2013	388	283	17% vs. 9.3%	47.2% (PD) vs. 39.9% (HC)

### 5.2. Vitamin D and Risk of Parkinson’s Disease

Several studies have reported an association between vitamin D and PD risk, even if results are sometimes conflicting (Table 2).

In 2010, Knekt et al. [22] published a longitudinal study based on The Mini-Finland Health Survey (Mini-Fin Study) to investigate the correlation between vitamin D levels and PD incidence, using the Cox proportional hazards model. Authors found that higher levels of vitamin D were associated with a lower risk of developing PD during a follow-up of 29 years (RR = 0.35; 95% CI 0.15–0.81, *p* = 0.006) [22]. Individuals who exhibited serum vitamin D concentrations greater than 50 nmol/L had a 65% lower risk of PD than those who had concentrations less than 25 nmol/L [22]. In 2017, Sleeman et al. [23] evaluated the association between vitamin D and newly diagnosed PD cases in a prospective observational study examining 145 PD patients compared with 94 healthy controls; authors reported that incident PD patients had significantly lower serum vitamin D concentrations at baseline than age-matched controls (44.1 ± 21.7 (mean ± sd) vs. 52.2 ± 22.1 nmol/L, respectively, *p* = 0.005); similarly, the PD group showed lower mean serum 25(OH)D concentrations than controls after a follow up of 18 months (44.2 ± 23.6 vs. 55.7 ± 28.8 nmol/L, respectively, *p* = 0.002). Another study has explored not only the correlation between vitamin D levels and PD risk, but also investigated the existence of possible differences in the neural network function in PD individuals according to vitamin D status. Lower levels of vitamin D were found in PD patients compared to healthy controls (23.60 ± 7.27 vs. 25.60 ± 5.78, *p* < 0.001); a high risk of PD was detected in individuals with vitamin D deficiency (25-(OH)-D < 20 ng/mL) (Odds ratio, OR = 2.319). PD patients with vitamin D deficiency had wider brain regions with altered fraction amplitude of low-frequency fluctuation than other PD groups and the corresponding healthy controls. Results indicated that low vitamin D levels in PD patients has an impact on neural network function and cerebral cortex [24].

Since dermal synthesis is the main source of vitamin D, some studies have focused on the relationship between outdoor work and the risk of PD. For instance, Danish men who worked outdoors were less likely to develop PD than those who worked indoors [25]. Odds ratios were 0.90 (95% CI 0.78 to 1.02), 0.86 (95% CI 0.75 to 0.99) and 0.72 (95% CI 0.63 to 0.82) respectively, in subjects with moderate, frequent, and maximal outdoor work compared to subjects who only worked indoors.

The authors postulated that outdoor workers may exhibit a lower risk for PD, presumably due to higher exposure to sunlight and therefore a decreased risk of hypovitaminosis D.

A French study [26], conducted on 69,010 incident PD patients, showed a low number of prescriptions of Parkinson’s medications in geographic areas with higher UV-B radiation; findings allowed to postulate that these areas are associated with a reduced incidence of PD. Some considerations should be done regarding the fact that sunlight exposure may differ between individuals regardless of appurtenance of the same geographic area. These studies reported an inverse correlation between vitamin D levels and sunlight exposure as a surrogate of vitamin D status in PD patients. This may presumably be related to reduced mobility and consequent outdoor activity. However, the role of diet and vitamin D intake on the risk for PD has been poorly explored in the literature.

Wang L. et al. [27] demonstrated an inverse association between PD risk and serum levels of all forms of vitamin D, including dietary 25-OH-D_2_, which does not depend on exposure to sunlight. These findings make it possible to hypothesize that a vitamin D related-PD risk may not simply be due to the lack of sunlight exposure. Other mechanisms may be implicated, such as gastrointestinal dysfunction, a common non-motor dysfunction in PD, which may compromise vitamin D_2_ absorption [28].

In addition to studies that demonstrated an association between vitamin D and PD, other studies did not support this relationship (Table 2).

A prospective study published by Shrestha and colleagues [29], who collected data from the Atherosclerosis Risk in Communities (ARIC) study [30], did not reveal any relationship between vitamin D and the incidence of PD. After a median follow-up period of 17 years, no significant association was detected between vitamin D status and PD risk, even though the authors noted an increased risk of PD when comparing vitamin D levels >30 ng/mL to levels <20 ng/mL. More recently, the Parkinson Associated Risk Syndrome (PARS) study aimed to assess vitamin D status in a cohort of patients at high risk for developing PD (subjects were considered at high risk if they displayed hyposmia according to the UPSIT scale and less than 80% age-expected putaminal dopamine transporter binding on DATscan). This study did not reveal any differences between total plasma vitamin D levels in high-risk patients compared with all other groups, raising suspicion about a link between vitamin D and PD risk [31]. Potential explanations for the inconsistent findings regarding the relationship between vitamin D and PD may include differences in geography, habits including diet, physical activity, and socioeconomics of the populations studied.

**Table 2 brainsci-12-00790-t002:** Association of Vitamin D status with PD risk, pro and cons evidences. Legend of abbreviations: PD (Parkinson’s Disease); HC (Healthy Controls). * Subjects from the Parkinson Associated Risk Syndrome (PARS) Study, a cohort of asymptomatic individuals, some of whom are at high risk for PD.

Association of Vitamin D Status with PD Risk						
Evidence Pro						
Ref.	Authorship	Type	Year	N° PD Patients	N° HC	Conclusions
[20]	Ding H et al.	cross-sectional and longitudinal case-control study	2013	388	283	Vitamin D3 was associated with PD (*p* values = 0.0034 and 0.047) n both univariate and multivariate analyses, respectively.
[22]	Knekt et al.	cohort study	2010	50	3123	Higher levels of vitamin D were associated with a lower risk of developing PD during a follow-up of 29 years (RR = 0.35; 95% CI 0.15–0.81, *p* = 0.006).
[23]	Sleeman et al.	prospective observational study	2017	145	94	Incident PD patients had significantly lower serum vitamin D concentrations at baseline than age-matched controls (44.1 ± 21.7 (mean ± sd) vs. 52.2 ± 22.1 nmol/L, respectively, *p* = 0.005); similarly, the PD group showed lower mean serum 25(OH)D concentrations than controls after a follow up of 18 months (44.2 ± 23.6 vs. 55.7 ± 28.8 nmol/L, respectively, *p* = 0.002).
[24]	Lv L et al.	cross-sectional study	2021	330	209	Lower levels of vitamin D were found in PD patients compared to healthy controls (23.60 ± 7.27 vs. 25.60 ± 5.78, *p* < 0.001) and a high risk of PD was detected in individuals with vitamin D deficiency (25-(OH)-D < 20 ng/mL) (Odds ratio, OR = 2.319).
[25]	Kenborg L et al.	case-control study	2011	3819	19,282	Odds ratios were 0.90 (95% CI 0.78 to 1.02), 0.86 (95% CI 0.75 to 0.99), and 0.72 (95% CI 0.63 to 0.82), respectively, in subjects with moderate, frequent, and maximal outdoor work compared to subjects who only worked indoors.
[26]	Kravietz A et al.	nationwide study	2017	69,010	//	Low number of prescriptions of Parkinson’s medications in geographic areas with higher UV-B radiation was detected.
[27]	Wang L et al.	case-control study	2014	478	431	Inverse association between PD risk and serum levels of all forms of vitamin D, including dietary 25-OH-D_2_, which does not depend on exposure to sunlight, was reported.
**Evidence cons**						
[29]	Shrestha S et al.	prospective cohort study	2016	67	12,695	No relationship between vitamin D and the incidence of PD was detected. After a median follow-up period of 17 years, no significant association was detected between vitamin D status and PD risk.
[31]	Fullard M et al.	cross-sectional study	2017	//	198 *	PARS study did not show any differences between total plasma vitamin D levels in high-risk patients compared with all other groups.

### 5.3. Vitamin D and the Pathophysiology of Parkinson’s Disease

Vitamin D acts on multiple biological processes, playing a fundamental role in various diseases ranging from skin diseases, cardiovascular disorders, autoimmune disorders, and neurological diseases, including PD [32]. Vitamin D has widespread effects on a large spectrum of systems and tissues, including the CNS, which goes beyond regulation of calcium homeostasis and bone metabolism [33]. Vitamin D acts on the CNS since it is a fat-soluble hormone able to cross the blood–brain barrier. Furthermore, the CNS can synthesize its own vitamin D, which yields local auto- or paracrine neurosteroid actions (at the local level). In the CNS, vitamin D has wide effects on cellular proliferation, differentiation, calcium signaling, neuroprotection, synaptogenesis, amyloid clearance, and prevention of neuronal death [33].

There is evidence that vitamin D is implicated in dopaminergic neurotransmission and cellular events such as neurogenesis and neurite development [34]. Vitamin D has been shown to increase the expression of the tyrosine hydroxylase enzyme in the chromaffin cells of the adrenal medulla, which display vitamin D receptors on the surface, increasing production of catecholamines [35]; furthermore, vitamin D seems to have a role both in the synthesis of dopamine in the CNS and in its storage [36]. It was also found that vitamin D attenuates neurotoxicity induced by 6-hydroxydopamine (a toxic compound) in rats, protecting against dopamine depletion in SNpc [37]. In vitro studies show that Vitamin D may upregulate the expression of the glial cell line-derived neurotrophic factor (GDNF), specifically in the striatum, suggesting its protective role in PD [38]. Moreover, vitamin D may display an important role in counteracting oxidative stress in the brain, since it reduces reactive oxygen species by different mechanisms, including PARP1 inhibition [39]. A recent study has reinforced the neuroprotective role of vitamin D which seems able to inhibit aggregation of α-synuclein, through the expression of calbindin-D28k, a calcium-binding protein [40].

The signaling of vitamin D is complex and includes different players. Upon vitamin D binding to the vitamin D receptor (VDR), localized on the cell membrane, VDR is internalized in the cytoplasm where it interacts with the retinoid X receptor. This complex then binds to vitamin D response elements of a large number of target genes, promoting their expression [41]. VDR is highly expressed not only in tissues and organs involved in metabolism of calcium, as reported above, but also in the CNS [41].

There is evidence to suggest that VDR may have a possible implication in the pathophysiology of PD. A recent report showed that VDR-knockout mice have a behavioral phenotype similar to human PD, exhibiting muscular and motor impairments, albeit with no detrimental effect on cognitive function [42].

Another key role could be played by 1α-hydroxylase, responsible for promoting the conversion of vitamin D into its active form. VDR and 1α-hydroxylase are expressed in different brain areas, not only in the rat, but also in humans, with a similar pattern. Dopaminergic neurons of the SNpc in particular, show a very high expression of VDR [43]; this area also displays a strong immunochemical presence of 1α-hydroxylase [43].

The significant expression of both the VDR and 1α-hydroxylase within the SNpc is of particular interest to elucidate the pathogenesis of PD. Studies have also found that the onset of VDR expression in the midbrain of rats seems to occur very early during embryonic development and coincides with the timing of development of dopaminergic neurons [44]. A recent study has analyzed the distribution and localization of 1α-hydroxylase and VDR in autoptic brain samples of PD patients [45]. The authors found that PD patients show a lower expression of 1α-hydroxylase in dopaminergic neurons than healthy counterparts. On the other hand, 1α-hydroxylase was overexpressed in astrocytes and specifically in cerebral structures implicated in PD (the dorsal motor nucleus of the vagus, SNpc, and frontal cortex). These astrocytes seem to be involved in the clearance of alpha-synuclein through autophagy [45].

### 5.4. Vitamin D Status and Parkinson’s Disease Severity

Various studies have investigated the association between serum vitamin D levels and the severity of PD motor symptoms. In the DATATOP study, a high rate of vitamin D deficiency and insufficiency was identified in PD patients at baseline. However, vitamin D concentrations did not decrease during the course of disease, suggesting that hypovitaminosis D is not correlated with motor severity [17]. A study conducted on a large cohort of PD patients from Oxford’s Parkinson’s Disease Centre Discovery investigated the role of vitamin D in disease progression (MDS- UPDRS I, II, III and Montreal Cognitive Assessment, MoCA, were used as outcomes) assessing serum vitamin D at baseline and after a mean follow-up period of 3.2 years. No association was found between baseline vitamin D levels and progression in motor and non-motor domains, except for a slight association with baseline routine daily activities (MDS-UPDRS II) [46]. Suzuki et al. found a significant, inverse correlation between vitamin D levels and total UPDRS (*p* = 0.004) and UPDRS III (*p* = 0.002) in a study including 137 PD participants [47]. The Harvard Biomarker Study, which analyzed data from 388 PD patients, gave similar results, showing that the total UPDRS score was inversely correlated with low 25-OH-D_3,_ as well as low total 25-OH-D levels (p = 0.0096 and 0.02 respectively). However, associations with the HY scale, another measure of disease severity, failed to reach statistical significance [20]. In addition, Sleeman et al. reported that baseline serum vitamin D concentrations were a predictor of motor severity (assessed by the UPDRS III score) at 36 months, even if no association with the risk of falling was detected [23]. Conversely, an observational pilot study including 40 PD patients reported a significant negative correlation between vitamin D levels and motor severity, assessed by the UPDRS III correlation (r = −0.33; *p* = 0.04), and a significant correlation with balance control (*p* < 0.05) was also detected [48].

### 5.5. Vitamin D Status and Non-Motor Symptoms

Few studies have assessed the relationship between vitamin D status and non-motor symptoms, such as cognition, depression, anxiety, insomnia, and orthostatic hypotension.

According to several investigations [49,50,51], vitamin D levels influence cognitive function [49,50,51]; however, to date, the impact of vitamin D on cognition in PD has been poorly explored. Underlying mechanisms include the influence of vitamin D on brain acetylcholine levels and the clearance of amyloid beta, both implicated in AD [51]. In CNS, cortical neurons and structures notably involved in processes of learning and memory, such as the hippocampus, are particularly rich in VDR and 24-hydroxylases, the enzyme responsible for catabolism of calcitriol [52]. In mice, a low post-natal vitamin D status compromises processes such as learning and hippocampal-dependent memory, suggesting as an adequate vitamin D intake is essential for an efficient hippocampal function [53]. Peterson el al. [54] conducted a study on 286 PD patients, divided into 2 groups: PD patients with dementia and PD patients without dementia. Using a multivariate Cox regression model, a positive correlation was found between scores obtained in several neuropsychiatric tests and vitamin D levels, only in the group without dementia. In particular, significant associations were found between vitamin D concentrations and verbal fluency and verbal memory (t = 4.31, *p* < 0.001 and t = 3.04, *p* = 0.0083). Conversely, no significant association was identified between cognition and vitamin D levels in the smaller subset with dementia (61 patients). Santangelo et al. [55] enrolled 60 untreated PD patients, monitoring vitamin D levels and cognitive function after a follow-up period of 24 months and 48 months. They found that lower levels of vitamin D at baseline were a predictor of mild cognitive impairment (MCI) occurrence after 48 months. Barichella et al. [56] conducted a cross-sectional observational study on a large sample of Italian PD patients (500 consecutive PD patients not taking vitamin D supplements compared with 100 controls) in order to obtain information regarding the potential role of vitamin D on clinical features. Vitamin D deficiency (<20 ng/mL) was found in 65.6% of PD patients, whereas 26.6% were found to have insufficient levels (20–30 ng/mL). Vitamin D levels were lower than those detected in controls (*p* = 0.006). Interestingly, patients with a deficient vitamin D status displayed more severe disease (*p* = 0.010) and worse clinical symptoms (total UPDRS (*p* = 0.040), UPDRS Part-III (*p* = 0.002), tremor *p* = 0.047, and bradykinesia *p* = 0.026). Moreover, greater impairment of global cognitive functions, investigated by MMSE (*p*= 0.001), was found in the vitamin D deficient cohort. A study by Gatto et al. has reported, among the various VDR polymorphisms, an association between FokI polymorphism and cognitive decline in PD patients, assessed through MMSE, during a follow-up period of 7.2 years [57]. Sleeman et al. [23] did not report any association between serum vitamin D and cognition during a follow-up period of 36 months. Globally, the majority of findings support the importance of detecting vitamin D deficiency in preventing cognitive impairment in PD patients. In addition to cognition, other studies have investigated the influence of vitamin D on other non-motor symptoms. Zhang et al. reported significantly lower serum vitamin D levels in PD patients compared to healthy controls (49.75 ± 14.11 vs. 43.40 ± 16.51, *p* < 0.001). PD patients reported higher frequency of insomnia (*p* = 0.015) and higher scores in scales assessing quality of sleep (Pittsburgh Sleep Quality Index), depression, and anxiety [58]. Peterson et al. [54] reported that vitamin D levels correlated with the depression score in the Geriatric Depression Scale (t = −3.08, *p* = 0.0083), suggesting that higher levels of vitamin D are associated with a better mood. In a study published by Jang et al., 55 PD participants were subdivided into two groups according to the presence or absence of orthostatic hypotension. The authors observed significantly lower serum 25-OH-D and calcitriol levels in patients with orthostatic hypotension than in the other group without orthostatic hypotension (19.99 ± 4.73 vs. 26.89 ± 4.82, *p* < 0.01 and 20.47 ± 6.79 vs. 35.50 ± 9.32, *p* < 0.01, respectively). Furthermore, 25-OH-D and calcitriol levels also showed significant negative correlations with changes in systolic (R = −0.44; *p* < 0.01) or diastolic blood pressure levels (R = −0.34; *p* < 0.05) [59]. Another study explored the relationship between vitamin D levels and nocturnal blood pressure changes, suggestive of autonomic dysfunction [60]. This study enrolled 35 PD patients, classified into three groups: “dippers” with a decline in mean nighttime blood pressure of >10% (normal finding), “non-dippers”, with a decline in mean nighttime blood pressure <10%, and “reverse dippers”, with an increased mean nighttime blood pressure, and failed to detect significant differences in vitamin D levels between the three groups.

### 5.6. Osteoporosis and Fracture Risk in Parkinson’s Disease

Reduced bone mass is a very common finding in PD, affecting up to 91% of women and 61% of men [61]. According to the World Health Organization, osteoporosis is defined as a bone mineral density less than 2.5 standard deviations, whereas osteopenia is a bone mineral density between 1.0 and 2.5 standard deviations below the average for age, race, and gender [62]. Reduced bone mass in PD patients seems to be caused mainly by reduced mobility through mechanisms similar to those observed in other neurological diseases. However, vitamin D deficiency along with other endocrine, nutritional and iatrogenic factors may play an important role in bone mass depletion [63]. In a large longitudinal study including 78,994 patients with osteoporosis and 78,994 controls (from the Korean National Health Insurance Service-Health Screening Cohort database from 2002 to 2015), the first group disclosed a higher incidence of PD (Incidence: 2.4:1000 vs. 1.4:1000 in the control group), suggesting osteoporosis may be a risk factor for PD [64]. A similar result was reported by another study conducted on 23,495 individuals with osteoporosis compared with 23,495 controls, which revealed in the first group a significant higher risk of PD (HR 1.31, 95% CI, 1.13–1.50, *p* < 0.001). A comparison between sexes showed that females with osteoporosis disclose a higher risk of PD (HR 1.50; 95% CI, 1.27–1.77, *p* < 0.001) than males (HR 1.23; 95% CI, 0.93–1.64, *p* = 0.15) [65]. Another cross-sectional study conducted on 54 PD patients compared with 59 healthy controls, not only found lower bone mass density in the PD group, but also detected a negative correlation between bone mass density in the hip and severity of PD (UPDRS II and UPDRS III scores as well as H&Y stage). These findings suggest that the degree of osteoporosis may reflect the progression of PD [66]. Some authors have also investigated the prevalence of osteoporosis and osteopenia in PD according to patients’ clinical phenotype. For this purpose, patients were divided into the two dominant clinical phenotypes: the postural instability gait difficulty-dominant type and the tremor-dominant type. Both groups showed a high prevalence of osteopenia, particularly in the femoral neck. However, the group with postural instability gait difficulty-dominant type group presented a high prevalence of osteoporosis, whereas no patients with osteoporosis were identified in the tremor-dominant type group and controls [67]. A meta-analysis including 23 studies on osteoporosis and PD showed that PD patients have a higher risk of osteoporosis (OR 2.61; 95% CI 1.69 to 4.03) and fractures compared with healthy controls (OR 2.28; 95% CI 1.83 to 2.83), with male patients having a lower risk of osteoporosis and osteopenia than female patients (OR 0.45; 95% CI 0.29 to 0.68). It has also been reported that PD patients have an increased risk of bone fractures compared with the general population. Bhattachary et al. [68] found a high prevalence of PD among a large population sample admitted to hospital for hip fractures in the United States between 1988 and 2007. The prevalence of PD was up to 4.48 times (95% CI: 4.46, 4.49) more than expected based on prevalence data, and was 4.02 (95% CI: 4.00, 4.03) when adjusted for gender and age. Similarly, Walker et al. [69] found that people affected by PD had an incidence of hip fracture higher than in people without PD living in North-East England, regardless of age. Another cross-sectional study enrolling 42 PD patients detected a higher degree of osteoporosis in PD subjects than in controls, more relevant in male sex (Z-score: M −3.8 ± 1.6, F −2.28 ± 0.92, *p* = 0.0006) [70].

A meta-analysis conducted on 31 independent studies of 564,947 participants pointed to a significant association between PD and the risk of hip fracture (HR 3.13, 95% CI 2.53–3.87) in both women 3.11 (2.51–3.86) and men 2.60 (2.19–3.09) [71].

### 5.7. Vitamin D Receptor Polymorphisms and Parkinson’s Disease

Some studies have investigated the relationship between VDR polymorphisms and PD, often with no conclusive results. Four classically typed VDR SNPs: rs731236 (TaqI); rs7975232 (ApaI); rs1544410 (BsmI); and rs2228570 (FokI), have been studied with regard to their possible association with PD. FokI seems to be the most implicated in the risk for PD. Suzuki et al. reported that PD patients with less motor severity present higher levels of vitamin D and disclose more commonly VDR FokI CC genotypes [47]. Interestingly, in PD patients with the FokI TT or CT genotypes, worsening of motor severity may be counteracted by vitamin D supplementation [47]. Similarly, the VDR FokI CC genotype was associated with milder forms of PD in Hungarian PD patients [72]. Furthermore, VDR FokI allele was associated with an increased risk of PD in a Chinese population [73].

A recent meta-analysis has analyzed the associations between all reported VDR polymorphisms and PD risk, showing a clear association between the FokI and susceptibility to PD in overall populations, particularly in Asian subgroups, whereas no relationship between TaqI, ApaI or BsmI, and PD was found [74].

Another meta-analysis on the association between VDR polymorphisms and PD concluded that not only FokI but also ApaI was significantly correlated with PD [75]. Moreover, a population-based, case-controlled study found that FokI seemed to be correlated with cognitive impairment in PD, as reported above [57]. The occurrence of FokI gene polymorphism may influence the risk, severity, and cognitive function of PD patients [76]. An association between VDR polymorphisms and PD risk depending on the level of UVR exposure was postulated. A study conducted in California, a region with high levels of UVR, did not find any association with FokI, whereas it did show associations with TaqI and ApaI loci [77].

## 6. Vitamin D and COVID-19 in Parkinson’s Disease

Vitamin D has recognized effects in the immune system and defense against infectious diseases, particularly viral respiratory infections [78]. It has been demonstrated that vitamin D reinforces mucosal integrity and counteracts the inflammatory response which contributes to acute respiratory distress syndrome and lung injury [79]. Recent studies published in the literature have investigated these immune-modulating properties of vitamin D against coronavirus disease 2019 (COVID-19), the severe acute respiratory syndrome caused by the SARS-CoV-2 virus, that has affected the world since the end of 2019, with controversial findings [80,81]. However, the mechanisms by which vitamin D modulates COVID-19 infection are not well determined. A possible protective effect of vitamin D might be due to its interaction with the angiotensin-converting enzyme 2 (ACE2) receptor, the host cell receptor responsible for mediating infection by SARS-CoV-2 [82]. Whereas SARS-CoV-2 downregulates the expression of ACE2, vitamin D seems to promote its expression [83]. High levels of ACE2 were found in dopaminergic neurons, which may imply more severe COVID-19 infection in PD patients [84]. Moreover, COVID-19 infection may determine the activation of human endogenous retroviruses (HERV), which may have a significant role in inflammation of the brain and neurodegenerative diseases, such as AD and PD. Other viral infections (HIV, herpes viruses) may act as co-players and induce HERV activation causing neuroinflammation and promoting neurodegeneration. Moreover, HERV activation may impair the response against viruses and other pathogens, preluding a more invasive behavior and a worse clinical outcome [85]. Two studies have found an inverse association between vitamin D serum concentrations and the prevalence and mortality of COVID-19 in the general population, suggesting a potential protective role of vitamin D in COVID-19 [86,87]. Conversely, low levels of vitamin D may be associated with a higher risk of infection, due to immune system impairment and a worsened outcome [88,89]. Cilia et al. [90] found worsened motor and non-motor symptoms in PD patients with COVID-19 when compared with a matched group of controls. Clinical deterioration was explained by infection-related mechanisms and impaired pharmacokinetics of dopaminergic therapy. A large case-controlled survey conducted in Lombardy, Italy, found that PD patients with COVID-19 were younger, more likely to suffer from chronic obstructive pulmonary disease, obese, and not taking vitamin D supplements than those without COVID-19. However, the risk of COVID-19 and mortality in PD patients did not differ from that in the general population [91]. Another study, published by Cereda et al. [92], did not find any correlation between vitamin D supplementation and a more favorable COVID-19 outcome in PD patients, raising concerns about the potential benefit of vitamin D in COVID-19.

## 7. Vitamin D and Cancer in Parkinson’s Disease

According to several evidences coming from many epidemiologic studies, vitamin D plays a protective role against cancer. In fact, it has been demonstrated an inverse correlation between vitamin D levels and risk of many cancers, including colon, breast, prostate, gastric, and other types of malignancies [93]. Beside epidemiological studies, cell-based research has underlined the antitumorigenic effects of vitamin D. Interestingly, many tumors are able to reduce calcitriol levels inside their cells, to counteract the protective effects of vitamin D. Indeed, vitamin D signalling and metabolism results are dysregulated in many cancers [93,94].

Its effects on inflammation (e.g., through inhibition of prostaglandins synthesis, p38 MAPK, and NF-kb signaling), DNA damage repair (e.g., through upregulation of p53, ATM, BRCA1 and other damage repair proteins), modulation of proliferation and differentiation (e.g., through upregulation of CDK inhibitors, activation of FOXO3/4, TGFB signaling), apoptosis, and autophagy are widely recognized. Among these several actions, calcitriol exerts a role in WNT–β-catenin signalling in colon cancer, in estrogens synthesis and signalling in estrogen receptor-positive (ER+) postmenopausal breast cancer, and, moreover, in androgen receptor (AR) signalling in prostate cancer [95].

Recently, the same molecular mechanisms through which vitamin D exerts its protective effects against tumorigenesis and cancer progression have been described as possibly involved in neurodegenerative diseases, including PD. Recent evidence has elucidated the role of MAPK signaling, WNT signaling, p53, NF-kB dysfunction, and other pathways in PD, suggesting the existence of a possible and intriguing link between vitamin D, PD, and cancer [96,97].

Expression of VDR is reduced in cancer cells, whereas enzymes involved in catabolism of vitamin D (CYP24A1) are aberrantly upregulated. Data regarding the expression of CYP27B1 in cancers are not univocal and dependent on several factors, including type of cancer and grade [98].

Studies regarding the association between PD and cancer are controversial [99,100]. Indeed, PD and cancer have a multifactorial etiopathogenesis which may explain in part the existence of contradictory results in this regard.

## 8. Vitamin D Supplementation in Parkinson’s Disease

The most significant information regarding the benefits of vitamin D supplementation in the general population derives from data generated by the 2017–2020 megatrials of vitamin D supplementation in largely vitamin D-repleted adults (VITAL, VIDA, D2d, DO-HEALTH) [101,102,103,104]. All these studies demonstrate that raising serum 25-OH-D concentrations into the high normal range (50–125 nmol/L or 20–50 ng/mL) does not create a significant benefit in clinical conditions, such as cardiovascular events, tumors, diabetes mellitus type 2, falls, or fractures. Therefore, vitamin D supplementation in (already) vitamin D-repleted individuals does not seem to provide protective effects [105]. PD patients often exhibit low levels of vitamin D [17,19,20,27], presumably correlated to an insufficient vitamin D intake, together with an insufficient intake of several micronutrients [21]. To date, few studies have investigated the effects of vitamin D supplementation on PD risk and disease progression, with conflicting findings. Some studies have shown that vitamin D supplementation and outdoor work were associated with a significantly reduced risk of PD [106,107]. Suzuki et al. conducted a randomized, placebo-controlled study examining the effect of supplementation with 1200 IU/day vitamin D on PD patients for 12 months. Serum vitamin D levels doubled in the group who took the supplement, whereas it did not increase in the placebo group. Motor severity, assessed through H&Y, remained unchanged in the supplemented group but worsened in the counterpart [108]. Another study assessed the effects of supplementation on dyskinesia in 120 PD patients assigned to supplementation with 1000 IU of vitamin D/day or assigned to placebo. No changes in dyskinesia and motor severity after 3 months were detected [109]. A longitudinal study which enrolled 1741 newly diagnosed PD subjects divided into four groups according to supplementation (no supplementation; multivitamin supplements; multivitamin plus vitamin D; only vitamin D) failed to identify any differences in disease progression across all groups after three years. In fact, no changes in the main outcome measures (Unified Parkinson’s Disease Rating Scale (UPDRS), routine daily activities, ADL, Parkinson’s disease questionnaire (PDQ), symbol digit modalities test (SDMT), or levodopa equivalent dose (LED) were detected at the end of follow-up [110].

A pilot randomized, double-blind intervention trial which aimed to measure the effects of 16 weeks of high dose vitamin D (10,000 IU/day) on balance in PD patients as measured by the Sensory Organization Test did not show improved balance, although a post hoc analysis identified evidence of a vitamin D effect on balance in the younger half of the cohort (mean age 60 years) [111]. Vitamin D supplementation seemed able to reduce stooped posture in patients with PD [112]. Barichella et al. evaluated the effects of a whey protein-based nutritional supplement enriched with vitamin D in patients with PD and parkinsonism, showing that consumption of this supplement was able to improve lower extremity function and preserve muscle mass [113].

No studies in the literature have investigated possible interactions between vitamin D supplementation and levodopa or other antiparkinsonian drugs.

## 9. Future Perspectives

It is not still clear if vitamin D supplementation may be used as a complementary therapy in PD patients or high-risk patients for developing the disease. All studies on vitamin D supplementation have provided conflicting findings. One of the main possible explanations could be that the amount of vitamin D used in clinical trials was not sufficient to provide a substantial benefit. Determining the optimal amount of vitamin D is crucial, since while low doses may be ineffective, high vitamin D doses may have toxic effects by disturbing intracellular calcium signaling (hypercalcemia). The choice of relevant endpoints like the risk of fractures, dementia, and hospitalization rates may be useful for this purpose. Further studies should consider factors such as ethnicity, latitude, age, BMI, diet, physical activity, and outdoor exposure, which may influence the optimal recommended dose of vitamin D. Whether vitamin D status or supplementation needs in idiopathic PD differ from genetic PD, where a lower vitamin D status may be detected, is a matter of ongoing debate [114].

Moreover, VDR polymorphisms should be considered since some genotypes have emerged as being likely to influence the risk and severity of PD, as well as the effect of vitamin D supplementation. To better elucidate the effects of vitamin D supplementation on the risk of PD, further randomized controlled supplementation trials with vitamin D should enroll patients with a high risk for PD (i.e., GBA carriers or patients with mutations in other susceptibility genes for PD, FokI CT/TT polymorphism, patients with RBD, patients with a strong PD family history). In addition, to clarify the relationship between vitamin D and PD and the effects of supplementation on motor progression, other studies should be conducted, stratifying PD patients according to different stages of the disease and clinical phenotype. This combined approach would be helpful to establish whether vitamin D may be therapeutically administered as a supplement to standard antiparkinsonian therapy and be effective in preventing disease.

In this review, we also analyzed the role of vitamin D in infectious diseases such as COVID-19. Moreover we explored the link between vitamin D and cancers in PD patients. In the literature, no investigations regarding the potential role of vitamin D in PD linked with cancer have been conducted. Vitamin D has a well-recognized protective role in a large number of cancers (e.g., breast cancer, gastrointestinal malignancies, melanoma), since it is implicated in tumors cell growth, inflammation, and immune surveillance [115,116,117,118]. However, the role of vitamin D in recent years has been widely discussed in malignant melanoma, whose occurrence has been reported with a higher frequency in PD, independently by dopaminergic therapy [93,117]. Further investigations are needed to clarify if PD patients who suffer from cancer may benefit from vitamin D, since a protective role of vitamin D supplementation in cancer prevention and/or progression has been described [117].

## 10. Conclusions

The relationship between low vitamin D status and PD is supported by several studies, from basic science to clinical applications.

Studies on animal models and, more recently, on the human brain emphasize the role of vitamin D on neuroprotection. Recently, Pignolo et al. [118] have extensively investigated the role of vitamin D in PD, describing its relevance in neuroprotection, its implication in all clinical aspects which characterize the disease, and suggesting a possible role of vitamin D supplementation as a possible therapeutic strategy.

In this review, we aimed to elucidate the correlation between vitamin D and PD from a molecular point of view to a clinical perspective.

We aimed to elucidate the pathophysiological link between vitamin D and PD.

Vitamin D depletion is responsible for the death of dopaminergic neurons [118]. On the other hand, vitamin D supplementation protects against dopamine depletion in SNpc in rats. The expression of VDR and 1α-hydroxylase in the brain suggests as vitamin D hydroxylation occurs in the brain, reinforcing the concept of the protective role of vitamin D against dopaminergic cells loss, and therefore PD. All these findings suggest the potential role of vitamin D as a biomarker in PD [36,37,41,118].

From a clinical perspective, a high prevalence of hypovitaminosis D in PD patients has been widely reported. Although the impact of hypovitaminosis D on PD risk and motor severity is fairly well elucidated, evidence regarding the association between hypovitaminosis D and cognitive impairment, as well as other non-motor symptoms, is less robust.

Moreover, we discussed the relevance of bone and calcium metabolism in PD. Hypovitaminosis D is a well-recognized risk factor of osteoporosis in PD. Bone fractures may reduce mobility and worsen motor performance, impacting heavily on the caregiver’s burden and lead to institutionalization. This complication reinforces the importance of vitamin D supplementation in PD patients who are often prone to fall.

Despite conflicting data, a growing body of evidence supports the beneficial role of vitamin D in PD, justifying the choice of supplementing all PD patients with suboptimal levels.

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
