# Peer review of "Vitamin D Status and Parkinson’s Disease"

_brainsci, 2022, doi:10.3390/brainsci12060790_

Round 1

Reviewer 1 Report

Dear Authors,

This review is on the amount of Vitamin D on the risk or the occurrence of Parkinson's disease.

The review is comprehensive and well complied. To begin with the Initial paragraphs provides an outline on the available sources of vitamin D,  its metabolism and deficiency which would make the reader aware of the background of the vitamin D. This would make it to easier to follow the rest of the manuscript. Enough number of articles have been identified and discussed in detail indicating the vitamin D insufficiency/deficiency in PD. Most recent review on PD with emphasis on vitamin D is also presented but one. Some contradicting data are also presented thus strengthening the impact of this article. 

Suggestions:

  1. Authors are requested to include the most recent review published in MDPI and discuss the importance of this review over that published by Pignolo.
  2. It would be great if a table can be included listing the articles used in this manuscript that does and doesn't show a correlation between the lower levels of vitamin D and PD.

Good luck!

Author Response

1.Authors are requested to include the most recent review published in MDPI and discuss the importance of this review over that published by Pignolo.

R1: We included in our manuscript the review recently published by Pignolo et al, who have extensively investigated the role of vitamin D in PD, describing its relevance in neuroprotection, its implication in all clinical aspects which characterize the disease and suggesting a possible role of vitamin D supplementation as a possible therapeutic strategy [106].

2.It would be great if a table can be included listing the articles used in this manuscript that does and doesn't show a correlation between the lower levels of vitamin D and PD.

R2: Thank you for your suggestion, a table could help to focus on the most relevant papers with does and does not show a correlation between hypovitaminosis and PD

Reviewer 2 Report

Though the issue concerning vitamin D and Parkinson's disease (PD) seems to be interest, this works generally lists work concerning both issues, without a sufficient critical view presented by authors. Moreover authors selectively evaluate the issues which may correspond with the association of vitamin D and PD e.g. authors mention SARS-CoV-2 however do not acknowledge that PD may be a disease partly impacted by neuroinflammation related to viral infection - HERV. Additionally features as deficiency of vitamin D in PD linked with cancers are also not described. 

Author Response

.Authors selectively evaluate the issues which may correspond with the association of vitamin D and PD e.g. authors mention SARS-CoV-2 however do not acknowledge that PD may be a disease partly impacted by neuroinflammation related to viral infection - HERV. 

R1: COVID 19 infection may determine the activation of human endogenous retroviruses (HERV) which may have a significant role in inflammation of the brain and neurodegenerative diseases, like AD and PD. Other viral infections (HIV, herpes viruses) may act as co-players and induce HERV activation, causing neuroinflammation and promoting neurodegeneration. Moreover, HERV activation may impair the response against viruses and other pathogens, preluding a more invasive behavior and a worse clinical outcome [85].

2.Additionally features as deficiency of vitamin D in PD linked with cancers are also not described. 

R2: Vitamin D has a well-recognized protective role in a large number of cancers (e.g. breast cancer, gastrointestinal malignancies), since it is implicated in tumors cell growth, inflammation and immune surveillance. Some studies have reported that hypovitaminosis D seems associated with an increased risk of developing certain types of cancers, suggesting a protective role of vitamin D supplementation in cancer prevention and/or progression. In literature no investigations regarding the potential role of vitamin D in PD linked with cancer have been conducted. However, the role of vitamin D in the last years has been widely discussed in malignant melanoma, whose occurrence has been reported with a higher frequency in PD, independently by dopaminergic therapy.  This may suggest the existence of an association between vitamin D and cancer in PD, and particularly between vitamin D and malignant melanoma, Further investigations are needed to clarify if PD patients who suffer from cancer may benefit from vitamin D. However, defining the existence of a such link is very challenging due to the heterogeneity of PD and cancer cases. In conclusion, despite conflicting data, the role of vitamin D in PD is widely described, justifying the choice of supplementing all PD patients with suboptimal levels.  The presence of other comorbidities in which vitamin D is implicated, such as cancer, is a finding which in some way may support this decision.

Round 2

Reviewer 2 Report

Authors have not sufficiently addressed my concerns e.g. vitamin D was presented in the context of cancers, however the elaboration on Parkinson's disease (PD) and its associations with cancer seems vague. Moreover the readership of the journal could expect a more extended discussion on the expected links of patomechanisms.

Author Response

Dear reviewer, 

the link between vitamin D, PD and cancer is very interesting, since never elucidated before. 

You can find our efforts in a brief but coincise paragraph. 

However this paragraph results beyond the scope of this manuscript. 

Kind regards, 

Federica Garrì
